# ResShift: Efficient Diffusion Model for Image Super-resolution by Residual Shifting

**Zongsheng Yue**    **Jianyi Wang**    **Chen Change Loy**
S-Lab, Nanyang Technological University
{zongsheng.yue,jianyi001,ccloy}@ntu.edu.sg

## Abstract

Diffusion-based image super-resolution (SR) methods are mainly limited by the low inference speed due to the requirements of hundreds or even thousands of sampling steps. Existing acceleration sampling techniques inevitably sacrifice performance to some extent, leading to over-blurry SR results. To address this issue, we propose a novel and efficient diffusion model for SR that significantly reduces the number of diffusion steps, thereby eliminating the need for post-acceleration during inference and its associated performance deterioration. Our method constructs a Markov chain that transfers between the high-resolution image and the low-resolution image by shifting the residual between them, substantially improving the transition efficiency. Additionally, an elaborate noise schedule is developed to flexibly control the shifting speed and the noise strength during the diffusion process. Extensive experiments demonstrate that the proposed method obtains superior or at least comparable performance to current state-of-the-art methods on both synthetic and real-world datasets, ***even only with 15 sampling steps***. Our code and model are available at https://github.com/zsyOAOA/ResShift.

## 1    Introduction

Image super-resolution (SR) is a fundamental problem in low-level vision, aiming at recovering the high-resolution (HR) image given the low-resolution (LR) one. This problem is severely ill-posed due to the complexity and unknown nature of degradation models in real-world scenarios. Recently, diffusion model [1, 2], a newly emerged generative model, has achieved unprecedented success in image generation  [3]. Furthermore, it has also demonstrated great potential in solving several downstream low-level vision tasks, including image editing [4, 5], image inpainting [6, 7], image colorization [8, 9]. There is also ongoing research exploring the potential of diffusion models to tackle the long-standing and challenging SR task.

One common approach [10, 11] involves inserting the LR image into the input of current diffusion model (e.g., DDPM [2]) and retraining the model from scratch on the training data for SR. Another popular way [7, 12, 13, 14] is to use an unconditional pre-trained diffusion model as a prior and modify its reverse path to generate the expected HR image. Unfortunately, both strategies inherit the Markov chain underlying DDPM, which can be inefficient in inference, often taking hundreds or even thousands of sampling steps. Although some acceleration techniques [15, 16, 17] have been developed to compress the sampling steps in inference, they inevitably lead to a significant drop in performance, resulting in over-smooth results as shown in Fig. 1, in which the DDIM [16] algorithm is employed to speed up the inference. Thus, there is a need to design a new diffusion model for SR that achieves both efficiency and performance, without sacrificing one for the other.

Let us revisit the diffusion model in the context of image generation. In the forward process, it builds up a Markov chain to gradually transform the observed data into a pre-specified prior distribution, typically a standard Gaussian distribution, over a large number of steps. Subsequently, image

37th Conference on Neural Information Processing Systems (NeurIPS 2023).

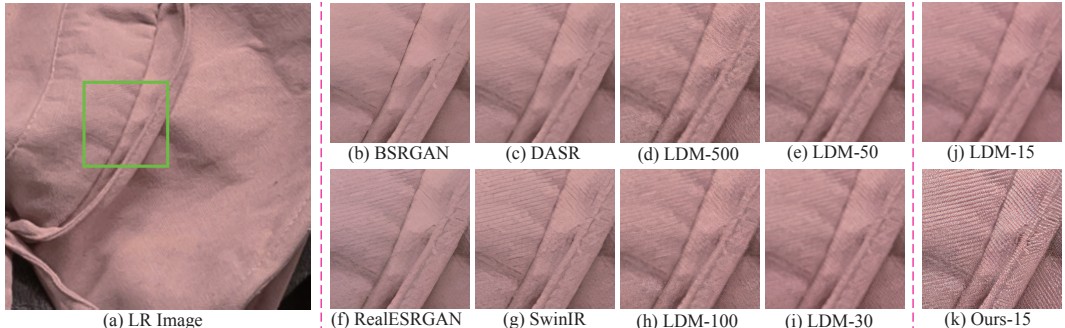

Figure 1: Qualitative comparisons on one typical real-world example of the proposed method and recent state of the arts, including BSRGAN [18], RealESRGAN [19], SwinIR [20], DASR [21], and LDM [11]. As for LDM and our method, we mark the number of sampling steps with the format of "LDM (or Ours)-A" for more intuitive visualization, where "A" is the number of sampling steps. Note that LDM contains 1000 diffusion steps in training and is accelerated to "A" steps using DDIM [16] during inference. Please zoom in for a better view.

generation can be achieved by sampling a noise map from the prior distribution and feeding it into the reverse path of the Markov chain. While the Gaussian prior is well-suited for the task of image generation, it may not be optimal for SR, where the LR image is available. In this paper, we argue that the reasonable diffusion model for SR should start from a prior distribution based on the LR image, enabling an iterative recovery of the HR image from its LR counterpart instead of Gaussian white noise. Additionally, such a design can reduce the number of diffusion steps required for sampling, thereby improving inference efficiency.

Following the aforementioned motivation, we propose an efficient diffusion model involving a shorter Markov chain for transitioning between the HR image and its corresponding LR one. The initial state of the Markov chain converges to an approximate distribution of the HR image, while the final state converges to an approximate distribution of the LR image. To achieve this, we carefully design a transition kernel that shifts the residual between them step by step. This approach is more efficient than existing diffusion-based SR methods since the residual information can be quickly transferred in dozens of steps. Moreover, our design also allows for an analytical and concise expression for the evidence lower bound, easing the induction of the optimization objective for training. Based on this constructed diffusion kernel, we further develop a highly flexible noise schedule that controls the shifting speed of the residual and the noise strength in each step. This schedule facilitates a fidelity-realism trade-off of the recovered results by tuning its hyper-parameters.

In summary, the main contributions of this work are as follows:

- We present an efficient diffusion model for SR, which renders an iterative sampling procedure from the LR image to the desirable HR one by shifting the residual between them during inference. Extensive experiments demonstrate the superiority of our approach in terms of efficiency, as it requires only 15 sampling steps to achieve appealing results, outperforming or at least being comparable to current diffusion-based SR methods that require a long sampling process. A preview of our recovered results compared with existing methods is shown in Fig. 1.

- We formulate a highly flexible noise schedule for the proposed diffusion model, enabling more precise control of the shifting of residual and noise levels during the transition.

## 2 Methodology

In this section, we present a diffusion model, *ResShift*, which is tailored for SR. For ease of presentation, the LR and HR images are denoted as $y_0$ and $x_0$, respectively. Furthermore, we assume $y_0$ and $x_0$ have identical spatial resolution, which can be easily achieved through pre-upsampling the LR image $y_0$ using nearest neighbor interpolation if necessary.

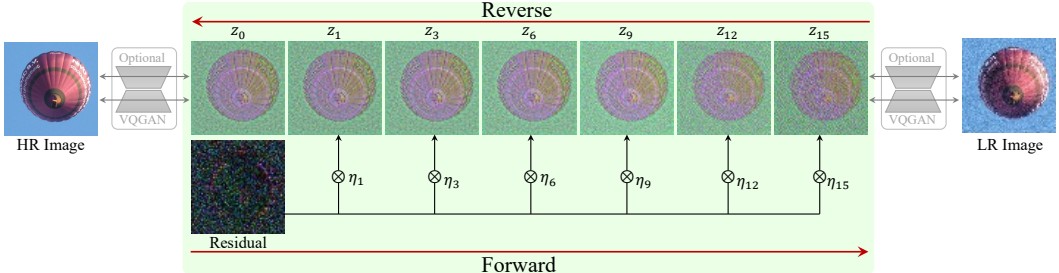

Figure 2: Overview of the proposed method. It builds up a Markov chain between the HR/LR image pair by shifting their residual.

## 2.1 Model Design

The iterative generation paradigm of diffusion models has proven highly effective at capturing complex distributions, inspiring us to approach the SR problem iteratively as well. Our proposed method constructs a Markov chain that serves as a bridge between the HR and LR images as shown in Fig. 2. This way, the SR task can be accomplished by reverse sampling from this Markov chain given any LR image. Next, we will detail the process of building such a Markov chain specifically for SR.

**Forward Process**. Let's denote the residual between the LR and HR images as $e_0$, i.e., $e_0 = y_0 - x_0$. Our core idea is to transit from $x_0$ to $y_0$ by gradually shifting their residual $e_0$ through a Markov chain with length $T$. A shifting sequence $\{\eta_t\}_{t=1}^{T}$ is first introduced, which monotonically increases with the timestep $t$ and satisfies $\eta_1 \to 0$ and $\eta_T \to 1$. The transition distribution is then formulated based on this shifting sequence as follows:

$$q(\boldsymbol{x}_t|\boldsymbol{x}_{t-1}, \boldsymbol{y}_0) = \mathcal{N}(\boldsymbol{x}_t; \boldsymbol{x}_{t-1} + \alpha_t \boldsymbol{e}_0, \kappa^2 \alpha_t \boldsymbol{I}), \; t = 1, 2, \cdots, T, \tag{1}$$

where $\alpha_t = \eta_t - \eta_{t-1}$ for $t > 1$ and $\alpha_1 = \eta_1$, $\kappa$ is a hyper-parameter controlling the noise variance, $\boldsymbol{I}$ is the identity matrix. Notably, we show that the marginal distribution at any timestep $t$ is analytically integrable, namely

$$q(\boldsymbol{x}_t|\boldsymbol{x}_0, \boldsymbol{y}_0) = \mathcal{N}(\boldsymbol{x}_t; \boldsymbol{x}_0 + \eta_t \boldsymbol{e}_0, \kappa^2 \eta_t \boldsymbol{I}), \; t = 1, 2, \cdots, T. \tag{2}$$

The design of the transition distribution presented in Eq. (1) is based on two primary principles. The first principle concerns the standard deviation, i.e., $\kappa\sqrt{\alpha_t}$, which aims to facilitate a smooth transition between $x_t$ and $x_{t-1}$. This is because the expected distance between $x_t$ and $x_{t-1}$ can be bounded by $\sqrt{\alpha_t}$, given that the image data falls within the range of $[0, 1]$, i.e.,

$$\max[(\boldsymbol{x}_0 + \eta_t \boldsymbol{e}_0) - (\boldsymbol{x}_0 + \eta_{t-1}\boldsymbol{e}_0)] = \max[\alpha_t \boldsymbol{e}_0] < \alpha_t < \sqrt{\alpha_t}, \tag{3}$$

where $\max[\cdot]$ represents the pixel-wise maximizing operation. The hyper-parameter $\kappa$ is introduced to increase the flexibility of this design. The second principle pertains to the mean parameter, i.e., $x_0 + \alpha_t e_0$, which induces the marginal distribution in Eq. (2). Furthermore, the marginal distributions of $x_1$ and $x_T$ converges to $\delta_{\boldsymbol{x}_0}(\cdot)$[1] and $\mathcal{N}(\cdot; \boldsymbol{y}_0, \kappa^2 \boldsymbol{I})$, which act as two approximate distributions for the HR image and the LR image, respectively. By constructing the Markov chain in such a thoughtful way, it is possible to handle the SR task by inversely sampling from it given the LR image $y_0$.

**Reverse Process**. The reverse process aims to estimate the posterior distribution $p(x_0|y_0)$ via the following formulation:

$$p(\boldsymbol{x}_0|\boldsymbol{y}_0) = \int p(\boldsymbol{x}_T|\boldsymbol{y}_0) \prod_{t=1}^{T} p_{\boldsymbol{\theta}}(\boldsymbol{x}_{t-1}|\boldsymbol{x}_t, \boldsymbol{y}_0) \mathrm{d}\boldsymbol{x}_{1:T}, \tag{4}$$

where $p(\boldsymbol{x}_T|\boldsymbol{y}_0) \approx \mathcal{N}(\boldsymbol{x}_T|\boldsymbol{y}_0, \kappa^2 \boldsymbol{I})$, $p_{\boldsymbol{\theta}}(\boldsymbol{x}_{t-1}|\boldsymbol{x}_t, \boldsymbol{y}_0)$ is the inverse transition kernel from $x_t$ to $x_{t-1}$ with a learnable parameter $\boldsymbol{\theta}$. Following most of the literature in diffusion model [1, 2, 8], we adopt the assumption of $p_{\boldsymbol{\theta}}(\boldsymbol{x}_{t-1}|\boldsymbol{x}_t, \boldsymbol{y}_0) = \mathcal{N}(\boldsymbol{x}_{t-1}; \boldsymbol{\mu}_{\boldsymbol{\theta}}(\boldsymbol{x}_t, \boldsymbol{y}_0, t), \boldsymbol{\Sigma}_{\boldsymbol{\theta}}(\boldsymbol{x}_t, \boldsymbol{y}_0, t))$. The optimization for $\boldsymbol{\theta}$ is achieved by minimizing the negative evidence lower bound, namely,

$$\min_{\boldsymbol{\theta}} \sum_{t} D_{\mathrm{KL}} \left[q(\boldsymbol{x}_{t-1}|\boldsymbol{x}_t, \boldsymbol{x}_0, \boldsymbol{y}_0) \| p_{\boldsymbol{\theta}}(\boldsymbol{x}_{t-1}|\boldsymbol{x}_t, \boldsymbol{y}_0)\right], \tag{5}$$

---

[1]$\delta_{\boldsymbol{\mu}}(\cdot)$ denotes the Dirac distribution centered at $\boldsymbol{\mu}$.

where $D_{\mathrm{KL}}[\cdot\|\cdot]$ denotes the Kullback-Leibler (KL) divergence. More mathematical details can be found in Sohl-Dickstein et al. [1] or Ho et al. [2].

Combining Eq. (1) and Eq. (2), the targeted distribution $q(\boldsymbol{x}_{t-1}|\boldsymbol{x}_t, \boldsymbol{x}_0, \boldsymbol{y}_0)$ in Eq. (5) can be rendered tractable and expressed in an explicit form given below:

$$q(\boldsymbol{x}_{t-1}|\boldsymbol{x}_t, \boldsymbol{x}_0, \boldsymbol{y}_0) = \mathcal{N}\left(\boldsymbol{x}_{t-1}\left|\frac{\eta_{t-1}}{\eta_t}\boldsymbol{x}_t + \frac{\alpha_t}{\eta_t}\boldsymbol{x}_0, \kappa^2\frac{\eta_{t-1}}{\eta_t}\alpha_t\boldsymbol{I}\right.\right). \tag{6}$$

The detailed calculation of this derivation is presented in the supplementary material. Considering that the variance parameter is independent of $\boldsymbol{x}_t$ and $\boldsymbol{y}_0$, we thus set $\boldsymbol{\Sigma_\theta}(\boldsymbol{x}_t, \boldsymbol{y}_0, t) = \kappa^2\frac{\eta_{t-1}}{\eta_t}\alpha_t\boldsymbol{I}$. As for the mean parameter $\boldsymbol{\mu_\theta}(\boldsymbol{x}_t, \boldsymbol{y}_0, t)$, it is reparameterized as follows:

$$\boldsymbol{\mu_\theta}(\boldsymbol{x}_t, \boldsymbol{y}_0, t) = \frac{\eta_{t-1}}{\eta_t}\boldsymbol{x}_t + \frac{\alpha_t}{\eta_t}f_{\boldsymbol{\theta}}(\boldsymbol{x}_t, \boldsymbol{y}_0, t), \tag{7}$$

where $f_{\boldsymbol{\theta}}$ is a deep neural network with parameter $\boldsymbol{\theta}$, aiming to predict $\boldsymbol{x}_0$. We explored different parameterization forms on $\boldsymbol{\mu_\theta}$ and found that Eq. (7) exhibits superior stability and performance.

Based on Eq. (7), we simplify the objective function in Eq. (5) as follows,

$$\min_{\boldsymbol{\theta}}\sum_t w_t\|f_{\boldsymbol{\theta}}(\boldsymbol{x}_t, \boldsymbol{y}_0, t) - \boldsymbol{x}_0\|_2^2, \tag{8}$$

where $w_t = \frac{\alpha_t}{2\kappa^2\eta_t\eta_{t-1}}$. In practice, we empirically find that the omission of weight $w_t$ results in an evident improvement in performance, which aligns with the conclusion in Ho et al. [2].

**Extension to Latent Space.** To alleviate the computational overhead in training, we move the aforementioned model into the latent space of VQGAN [22], where the original image is compressed by a factor of four in spatial dimensions. This does not require any modifications on our model other than substituting $\boldsymbol{x_0}$ and $\boldsymbol{y}_0$ with their latent codes. An intuitive illustration is shown in Fig. 2.

## 2.2 Noise Schedule

The proposed method employs a hyper-parameter $\kappa$ and a shifting sequence $\{\eta_t\}_{t=1}^T$ to determine the noise schedule in the diffusion process. Specifically, the hyper-parameter $\kappa$ regulates the overall noise intensity during the transition, and its impact on performance is empirically discussed in Sec. 4.2. The subsequent exposition mainly revolves around the construction of the shifting sequence $\{\eta_t\}_{t=1}^T$.

Equation (2) implies that the noise level in state $\boldsymbol{x}_t$ is proportional to $\sqrt{\eta_t}$ with a scaling factor $\kappa$. This observation motivates us to focus on designing $\sqrt{\eta_t}$ instead of $\eta_t$. Song and Ermon [23] show that $\kappa\sqrt{\eta_1}$ should be sufficiently small (e.g., 0.04 in LDM [11]) to ensure that $q(\boldsymbol{x}_1|\boldsymbol{x}_0, \boldsymbol{y}_0) \approx q(\boldsymbol{x}_0)$. Combining with the additional constraint of $\eta_1 \to 0$, we set $\eta_1$ to be the minimum value between $(0.04/\kappa)^2$ and 0.001. For the final step $T$, we set $\eta_T$ as 0.999 ensuring $\eta_T \to 1$. For the intermediate timesteps, i.e., $t \in [2, T-1]$, we propose a non-uniform geometric schedule for $\sqrt{\eta_t}$ as follows:

$$\sqrt{\eta_t} = \sqrt{\eta_1} \times b_0^{\beta_t}, \; t = 2, \cdots, T-1, \tag{9}$$

where

$$\beta_t = \left(\frac{t-1}{T-1}\right)^p \times (T-1), \; b_0 = \exp\left[\frac{1}{2(T-1)}\log\frac{\eta_T}{\eta_1}\right]. \tag{10}$$

Note that the choice of $\beta_t$ and $b_0$ is based on the assumption of $\beta_1 = 0$, $\beta_T = T-1$, and $\sqrt{\eta_T} = \sqrt{\eta_1} \times b_0^{T-1}$. The hyper-parameter $p$ controls the growth rate of $\sqrt{\eta_t}$ as shown in Fig. 3(h).

The proposed noise schedule exhibits high flexibility in three key aspects. First, for small values of $\kappa$, the final state $\boldsymbol{x}_T$ converges to a perturbation around the LR image as depicted in Fig. 3(c)-(d). Compared to the corruption ended at Gaussian noise, this design considerably shortens the length of the Markov chain, thereby improving the inference efficiency. Second, the hyper-parameter $p$ provides precise control over the shifting speed, enabling a fidelity-realism trade-off in the SR results as analyzed in Sec. 4.2. Third, by setting $\kappa = 40$ and $p = 0.8$, our method achieves a diffusion process remarkably similar to LDM [11]. This is clearly demonstrated by the visual results during the diffusion process presented in Fig. 3(e)-(f), and further supported by the comparisons on the relative noise strength as shown in Fig. 3(g).

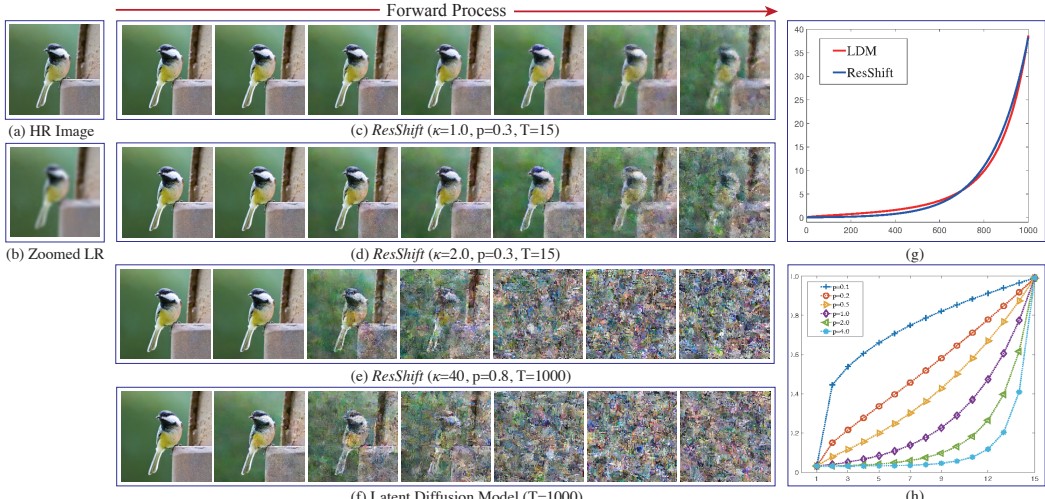

Figure 3: Illustration of the proposed noise schedule. (a) HR image. (b) Zoomed LR image. (c)-(d) Diffused images of *ResShift* in timesteps of 1, 3, 5, 7, 9, 12, and 15 under different values of $\kappa$ by fixing $p = 0.3$ and $T = 15$. (e)-(f) Diffused images of *ResShift* with a specified configuration of $\kappa = 40, p = 0.8, T = 1000$ and LDM [11] in timesteps of 100, 200, 400, 600, 800, 900, and 1000. (g) The relative noise intensity (vertical axes, measured by $\sqrt{1/\lambda_{\text{snr}}}$, where $\lambda_{\text{snr}}$ denotes the signal-to-noise ratio) of the schedules in (d) and (e) w.r.t. the timesteps (horizontal axes). (h) The shifting speed $\sqrt{\eta_t}$ (vertical axes) w.r.t. to the timesteps (horizontal axes) across various configurations of $p$. Note that the diffusion processes in this figure are implemented in the latent space, but we display the intermediate results after decoding back to the image space for the purpose of easy visualization.

## 3 Related Work

**Diffusion Model**. Inspired by the non-equilibrium statistical physics, Sohl-Dickstein et al. [1] firstly proposed the diffusion model to fit complex distributions. Ho et al. [2] established a novel connection between the diffusion model and the denoising scoring matching. Later, Song et al. [8] proposed a unified framework to formulate the diffusion model from the perspective of the stochastic differential equation (SDE). Attributed to its robust theoretical foundation, the diffusion model has achieved impressive success in the generation of images [3, 11], audio [24], graph [25] and shapes [26].

**Image Super-Resolution**. Traditional image SR methods primarily focus on designing more rational image priors based on our subjective knowledge, such as non-local similarity [27], low-rankness [28], sparsity [29, 30], and so on. With the development of deep learning (DL), Dong et al. [31] proposed the seminal work SRCNN to solve the SR task using a deep neural network. Then DL-based SR methods rapidly dominated the research field. Various SR technologies were explored from different perspectives, including network architecture [32, 33, 34, 35], image prior [36, 37, 38, 39], deep unfolding [40, 41, 42], degradation model [18, 19, 43, 44].

Recently, some works have investigated the application of diffusion models in SR. A prevalent approach is to concatenate the LR image with the noise in each step and retrain the diffusion model from scratch [10, 11, 45]. Another popular way is to utilize an unconditional pre-trained diffusion model as a prior and incorporate additional constraints to guide the reverse process [7, 12, 13, 46]. Both strategies often require hundreds or thousands of sampling steps to generate a realistic HR image. While several acceleration algorithms [15, 16, 17] have been proposed, they typically sacrifice the performance and result in blurry outputs. This work designs a more efficient diffusion model that overcomes this trade-off between efficiency and performance, as detailed in Sec. 2.

**Remark**. Several parallel works [47, 48, 49] also exploit such an iterative restoration paradigm in SR. Despite a similar motivation, our work and others have adopted different mathematical formulations to achieve this goal. Delbracio and Milanfar [47] employed the Inversion by Direct Iteration (InDI) to model this process, while Luo et al. [48] and Liu et al. [49] attempted to formulate it as a SDE. In this paper, we design a discrete Markov chain to depict the transition between the HR and LR images, offering a more intuitive and efficient solution to this problem.

Table 1: Performance comparison of *ResShift* on the *ImageNet-Test* under different configurations.

| Configurations | | | Metrics | | | | |
|---|---|---|---|---|---|---|---|
| $T$ | $p$ | $\kappa$ | PSNR↑ | SSIM↑ | LPIPS↓ | CLIPIQA↑ | MUSIQ↑ |
| 10 | | | 25.20 | 0.6828 | 0.2517 | 0.5492 | 50.6617 |
| 15 | | | 25.01 | 0.6769 | 0.2312 | 0.5922 | 53.6596 |
| 30 | 0.3 | 2.0 | 24.52 | 0.6585 | 0.2253 | 0.6273 | 55.7904 |
| 40 | | | 24.29 | 0.6513 | 0.2225 | 0.6468 | 56.8482 |
| 50 | | | 24.22 | 0.6483 | 0.2212 | 0.6489 | 56.8463 |
| | 0.3 | | 25.01 | 0.6769 | 0.2312 | 0.5922 | 53.6596 |
| | 0.5 | | 25.05 | 0.6745 | 0.2387 | 0.5816 | 52.4475 |
| 15 | 1.0 | 2.0 | 25.12 | 0.6780 | 0.2613 | 0.5314 | 48.4964 |
| | 2.0 | | 25.32 | 0.6827 | 0.3050 | 0.4601 | 43.3060 |
| | 3.0 | | 25.39 | 0.5813 | 0.3432 | 0.4041 | 38.5324 |
| | | 0.5 | 24.90 | 0.6709 | 0.2437 | 0.5700 | 50.6101 |
| | | 1.0 | 24.84 | 0.6699 | 0.2354 | 0.5914 | 52.9933 |
| 15 | 0.3 | 2.0 | 25.01 | 0.6769 | 0.2312 | 0.5922 | 53.6596 |
| | | 8.0 | 25.31 | 0.6858 | 0.2592 | 0.5231 | 49.3182 |
| | | 16.0 | 24.46 | 0.6891 | 0.2772 | 0.4898 | 46.9794 |

Figure 4: Qualitative comparisons of *ResShift* under different combinations of $(T, p, \kappa)$. For example, "(15, 0.3, 2.0)" represents the recovered result with $T = 15$, $p = 0.3$, and $\kappa = 2.0$. Please zoom in for a better view.

## 4 Experiments

This section presents an empirical analysis of the proposed *ResShift* and provides extensive experimental results to verify its effectiveness on one synthetic dataset and three real-world datasets. Following [18, 19], our investigation specifically focuses on the more challenging $\times 4$ SR task. Due to page limitation, some experimental results are put in the supplementary material.

### 4.1 Experimental Setup

**Training Details**. HR images with a resolution of $256 \times 256$ in our training data are randomly cropped from the training set of ImageNet [50] following LDM [11]. We synthesize the LR images using the degradation pipeline of RealESRGAN [19]. The Adam [51] algorithm with the default settings of PyTorch [52] and a mini-batch size of 64 is used to train *ResShift*. During training, we use a fixed learning rate of 5e-5 and update the weight parameters for 500K iterations. As for the network architecture, we employ the UNet structure in DDPM [2]. To increase the robustness of

Table 2: Efficiency and performance comparisons of *ResShift* to other methods on the dataset of *ImageNet-Test*. "LDM-A" represents the results achieved by accelerated the sampling steps of LDM [11] to "A". Running time is tested on NVIDIA Tesla V100 GPU on the x4 (64→ 256) SR task.

| Metrics | Methods | | | | | | |
|---|---|---|---|---|---|---|---|
| | BSRGAN | RealESRGAN | SwinIR | LDM-15 | LDM-30 | LDM-100 | *ResShift* |
| PSNR↑ | 24.42 | 24.04 | 23.99 | 24.89 | 24.49 | 23.90 | 25.01 |
| LPIPS↓ | 0.259 | 0.254 | 0.238 | 0.269 | 0.248 | 0.244 | 0.231 |
| CLIPIQA↑ | 0.581 | 0.523 | 0.564 | 0.512 | 0.572 | 0.620 | 0.592 |
| Runtime (s) | 0.012 | 0.013 | 0.046 | 0.102 | 0.184 | 0.413 | 0.105 |
| # Parameters (M) | 16.70 | 16.70 | 28.01 | 113.60 | | | 118.59 |

*ResShift* to arbitrary image resolution, we replace the self-attention layer in UNet with the Swin Transformer [53] block.

**Testing Datasets**. We synthesize a testing dataset that contains 3000 images randomly selected from the validation set of ImageNet [50] based on the commonly-used degradation model, i.e., $y = (x * k) \downarrow + n$, where $k$ is the blurring kernel, $n$ is the noise, $y$ and $x$ denote the LR image and HR image, respectively. To comprehensively evaluate the performance of *ResShift*, we consider more complicated types of blurring kernels, downsampling operators, and noise types. The detailed settings on them can be found in the supplementary material. It should be noted that we selected the HR images from ImageNet [50] instead of the prevailing datasets in SR such as *Set5* [54], *Set14* [55], and *Urban100* [56]. The rationale behind this setting is rooted in the fact that these datasets only contain very few source images, which fails to thoroughly evaluate the performance of various methods under different degradation types. We name this dataset as *ImageNet-Test* for convenience.

Two real-world datasets are adopted to evaluate the efficacy of *ResShift*. The first is *RealSR* [57], containing 100 real images captured by Canon 5D3 and Nikon D810 cameras. Additionally, we collect another real-world dataset named *RealSet65*. It comprises 35 LR images widely used in recent literature [19, 58, 59, 60, 61]. The remaining 30 images were obtained from the internet by ourselves.

**Compared Methods**. We evaluate the effectiveness of *ResShift* in comparison to seven recent SR methods, namely ESRGAN [62], RealSR-JPEG [63], BSRGAN [18], RealESRGAN [19], SwinIR [20], DASR [21], and LDM [11]. Note that LDM is a diffusion-based method with 1,000 diffusion steps. For a fair comparison, we accelerate LDM to the same number of steps with *ResShift* using DDIM [16] and denote it as "LDM-A", where "A" indicates the number of inference steps. The hyper-parameter $\eta$ in DDIM is set to be 1 as this value yields the most realistic recovered images.

**Metrics**. The performance of various methods was assessed using five metrics, including PSNR, SSIM [64], LPIPS [65], MUSIQ [66], and CLIPIQA [67]. It is worth noting that the latter two are non-reference metrics specifically designed to assess the realism of images. CLIPIQA, in particular, leverages the CLIP [68] model that is pre-trained on a massive dataset (i.e., Laion400M [69]) and thus demonstrates strong generalization ability. On the real-world datasets, we mainly rely on CLIPIQA and MUSIQ as evaluation metrics to compare the performance of different methods.

## 4.2 Model Analysis

We analyze the performance of *ResShift* under different settings on the number of diffusion steps $T$ and the hyper-parameters $p$ in Eq. (10) and $\kappa$ in Eq. (1).

**Diffusion Steps $T$ and Hyper-parameter $p$**. The proposed transition distribution in Eq. (1) significantly reduces the diffusion steps $T$ in the Markov chain. The hyper-parameter $p$ allows for flexible control over the speed of residual shifting during the transition. Table 1 summarizes the performance of *ResShift* on *ImageNet-Test* under different configurations of $T$ and $p$. We can see that both of $T$ and $p$ render a trade-off between the fidelity, measured by the reference metrics such as PSNR, SSIM, and LPIPS, and the realism, measured by the non-reference metrics, including CLIPIQA and MUSIQ, of the super-resolved results. Taking $p$ as an example, when it increases, the reference metrics improve while the non-reference metrics deteriorate. Furthermore, the visual comparison in Fig. 4 shows that a large value of $p$ will suppress the model's ability to hallucinate more image details and result in blurry outputs.

**Hyper-parameter $\kappa$**. Equation (2) reveals that $\kappa$ dominates the noise strength in state $x_t$. We report the influence of $\kappa$ to the performance of *ResShift* in Table 1. Combining with the visualization in

Table 3: Quantitative results of different methods on the dataset of *ImageNet-Test*. The best and second best results are highlighted in **bold** and underline.

| Methods | Metrics | | | | |
|---|---|---|---|---|---|
| | PSNR↑ | SSIM↑ | LPIPS↓ | CLIPIQA↑ | MUSIQ↑ |
| ESRGAN [62] | 20.67 | 0.448 | 0.485 | 0.451 | 43.615 |
| RealSR-JPEG [63] | 23.11 | 0.591 | 0.326 | 0.537 | 46.981 |
| BSRGAN [18] | 24.42 | 0.659 | 0.259 | 0.581 | **54.697** |
| SwinIR [20] | 23.99 | 0.667 | 0.238 | 0.564 | 53.790 |
| RealESRGAN [19] | 24.04 | 0.665 | 0.254 | 0.523 | 52.538 |
| DASR [21] | 24.75 | 0.675 | 0.250 | 0.536 | 48.337 |
| LDM-15 [11] | 24.89 | 0.670 | 0.269 | 0.512 | 46.419 |
| *ResShift* | **25.01** | **0.677** | **0.231** | **0.592** | 53.660 |

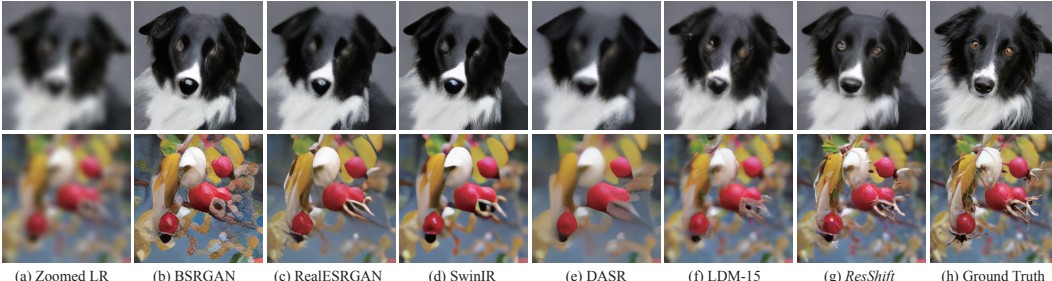

| (a) Zoomed LR | (b) BSRGAN | (c) RealESRGAN | (d) SwinIR | (e) DASR | (f) LDM-15 | (g) *ResShift* | (h) Ground Truth |

Figure 5: Qualitative comparisons of different methods on two synthetic examples of the *ImageNet-Test* dataset. Please zoom in for a better view.

Fig. 4, we can find that excessively large or small values of $\kappa$ will smooth the recovered results, regardless of their favorable metrics of PSNR and SSIM. When $\kappa$ is in the range of $[1.0, 2.0]$, our method achieves the most realistic quality indicated by CLIPIQA and MUSIQ, which is more desirable in real applications. We thus set $\kappa$ to be 2.0 in this work.

**Efficiency Comparison**. To improve inference efficiency, it is desirable to limit the number of diffusion steps $T$. However, this causes a decrease in the realism of the restored HR images. To compromise, the hyper-parameter $p$ can be set to a relatively small value. Therefore, we set $T = 15$ and $p = 0.3$, and yield our model named *ResShift*. Table 2 presents the efficiency and performance comparisons of *ResShift* to the state-of-the-art (SotA) approach LDM [11] and three other GAN-based methodologies on *ImageNet-Test* dataset. It is evident from the results that the proposed *ResShift* surpasses LDM [11] in terms of PSNR and LPIPS [65], and demonstrates a remarkable fourfold enhancement in computational efficiency when compared to LDM-100. Despite showing considerable potential in mitigating the efficiency bottleneck of the diffusion-based SR approaches, *ResShift* still lags behind current GAN-based methods in speed due to its iterative sampling mechanism. Therefore, it remains imperative to explore further optimizations of the proposed method to address this limitation, which we leave in our future work.

**Perception-Distortion Trade-off.** There exists a well-known phenomenon called perception-distortion trade-off [70] in the field of SR. In particular, the augmentation of the generative capability of a restoration model, such as elevating the sampling steps for a diffusion-based method or amplifying the weight of the adversarial loss for a GAN-based method, will result in a deterioration in fidelity preservation while concurrently enhancing the authenticity of restored images. That is mainly because the restoration model with powerful generation capability tends to hallucinate more high-frequency image structures, thereby deviating from the underlying ground truth. To facilitate a comprehensive comparison between our *ResShift* and current SotA diffusion-based method LDM, we plotted the perception-distortion curves of them in Fig. 7, wherein the perception and distortion are mea-

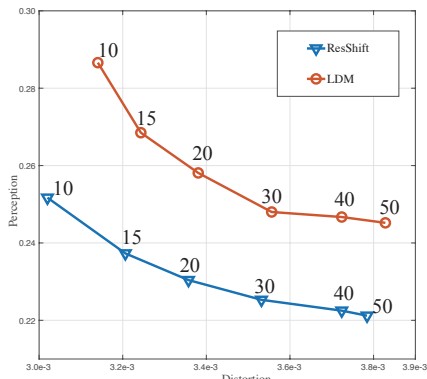

Figure 7: Perception-distortion trade-off of *ResShift* and LDM. The vertical and horizontal axes represent the strength of the perception and distortion, measured by LPIPS and MSE, respectively.

Table 4: Quantitative results of different methods on two real-world datasets. The best and second best results are highlighted in **bold** and underline.

| Methods | Datasets | | | |
| | RealSR | | RealSet65 | |
| | CLIPIQA↑ | MUSIQ↑ | CLIPIQA↑ | MUSIQ↑ |
|---|---|---|---|---|
| ESRGAN [62] | 0.2362 | 29.048 | 0.3739 | 42.369 |
| RealSR-JPEG [63] | 0.3615 | 36.076 | 0.5282 | 50.539 |
| BSRGAN [18] | 0.5439 | **63.586** | 0.6163 | **65.582** |
| SwinIR [20] | 0.4654 | 59.636 | 0.5782 | 63.822 |
| RealESRGAN [19] | 0.4898 | 59.678 | 0.5995 | 63.220 |
| DASR [21] | 0.3629 | 45.825 | 0.4965 | 55.708 |
| LDM-15 [11] | 0.3836 | 49.317 | 0.4274 | 47.488 |
| *ResShift* | **0.5958** | 59.873 | **0.6537** | 61.330 |

sured by LPIPS and mean square-error (MSE), respectively. This plot reflects the perception quality and the reconstruction fidelity of *ResShift* and LDM across varying numbers of diffusion steps, i.e., 10, 15, 20, 30, 40, and 50. As can be observed, the perception-distortion curve of our *ResShift* consistently resides beneath that of the LDM, indicating its superior capacity in balancing perception and distortion.

## 4.3    Evaluation on Synthetic Data

We present a comparative analysis of the proposed method with recent SotA approaches on the *ImageNet-Test* dataset, as summarized in Table 3 and Fig. 5. Based on this evaluation, several significant conclusions can be drawn as follows: i) *ResShift* exhibits superior or at least comparable performance across all five metrics, affirming the effectiveness and superiority of the proposed method. ii) The notably higher PSNR ans SSIM values attained by *ResShift* indicate its capacity to better preserve fidelity to ground truth images. This advantage primarily arises from our well-designed diffusion model, which starts from a subtle disturbance of the LR image, rather than the conventional assumption of white Gaussian noise in LDM. iii) Considering the metrics of LPIPS and CLIPIQA, which gauge the perceptual quality and realism of the recovered image, *ResShift* also demonstrates evident superiority over existing methods. Furthermore, in terms of MUSIQ, our approach achieves comparable performance with recent SotA methods. In summary, the proposed *ResShift* exhibits remarkable capabilities in generating more realistic results while preserving fidelity. This is of paramount importance for the task of SR.

## 4.4    Evaluation on Real-World Data

Table 4 lists the comparative evaluation using CLIPIQA [67] and MUSIQ [66] of various methods on two real-world datasets. Note that CLIPIQA, benefiting from the powerful representative capability inherited from CLIP, performs stably and robustly in assessing the perceptional quality of natural images. The results in Table 4 show that the proposed *ResShift* evidently surpasses existing methods in CLIPIQA, meaning that the restored outputs of *ResShift* better align with human visual and perceptive systems. In the case of MUSIQ evaluation, *ResShift* achieves the competitive performance when compared to current SotA methods, namely BSRGAN [18], SwinIR [20], and RealESRGAN [19]. Collectively, our method shows promising capability in addressing the real-world SR problem.

We display four real-world examples in Fig. 6. More examples can be found in the supplementary material. We consider diverse scenarios, including comic, text, face, and natural images to ensure a comprehensive evaluation. A noticeable observation is that *ResShift* produces more naturalistic image structures, as evidenced by the patterns on the beam in the third example and the eyes of a person in the fourth example. We note that the recovered results of LDM are excessively smooth when compressing the inference steps to match with the proposed *ResShift*, specifically utilizing 15 steps, largely deviating from the training procedure's 1,000 steps. Even though other GAN-based methods may also succeed in hallucinating plausible structures to some extent, they are often accompanied by obvious artifacts.

## 5    Conclusion

In this work, we have introduced an efficient diffusion model named *ResShift* for SR. Unlike existing diffusion-based SR methods that require a large number of iterations to achieve satisfactory results,

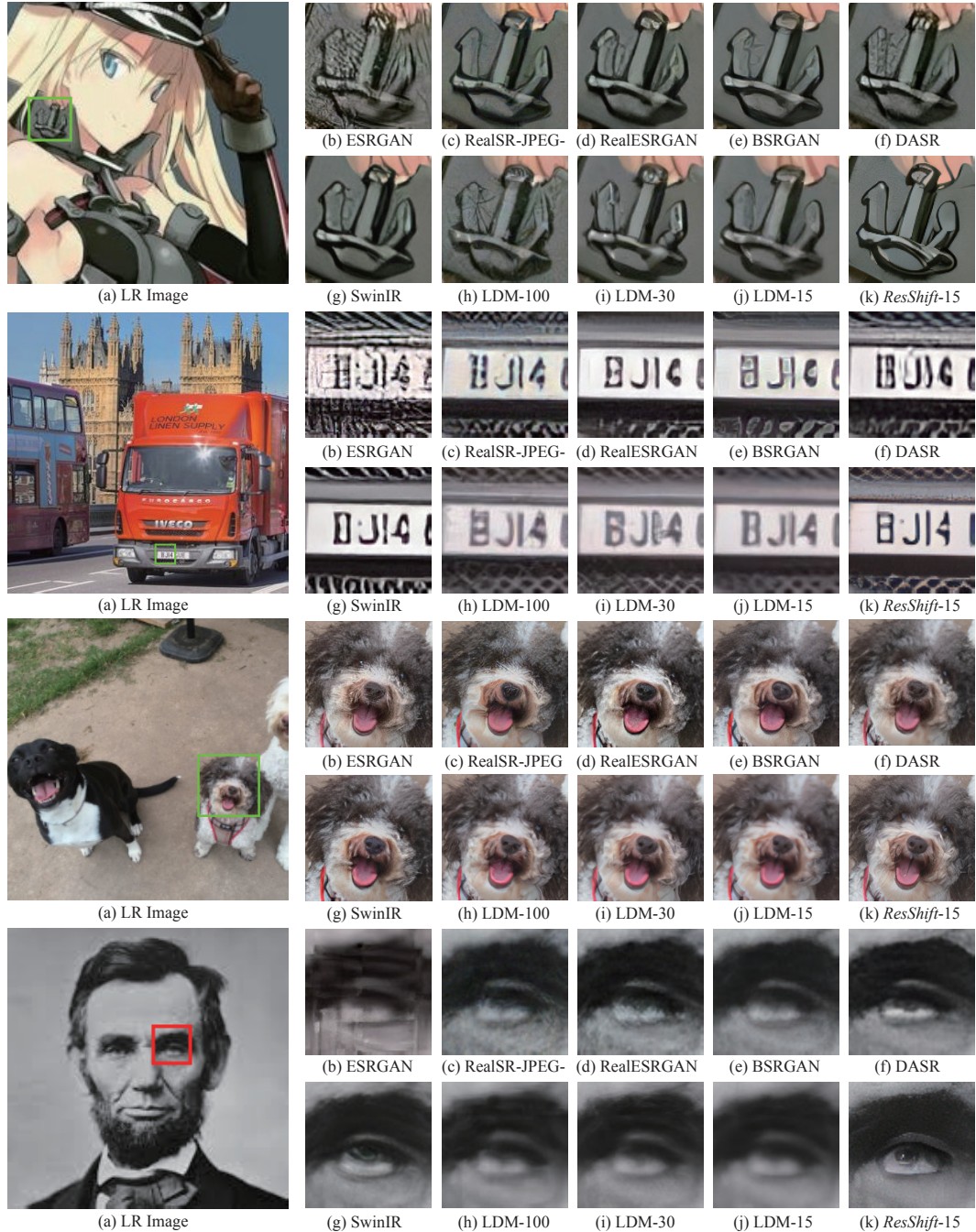

Figure 6: Qualitative comparisons on four real-world examples. Please zoom in for a better view.

our proposed method constructs a diffusion model with only 15 sampling steps, thereby significantly improving inference efficiency. The core idea is to corrupt the HR image toward the LR image instead of the Gaussian white noise, which can effectively cut off the length of the diffusion model. Extensive experiments on both synthetic and real-world datasets have demonstrated the superiority of our proposed method. We believe that our work will pave the way for the development of more efficient and effective diffusion models to address the SR problem.

**Acknowledgement.** This study is supported under the RIE2020 Industry Alignment Fund – Industry Collaboration Projects (IAF-ICP) Funding Initiative, as well as cash and in-kind contribution from the industry partner(s).

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
