# Supplementary Material of "ResShift: Efficient Diffusion Model for Image Super-resolution by Residual Shifting"

**Zongsheng Yue**    **Jianyi Wang**    **Chen Change Loy**
S-Lab, Nanyang Technological University
{zongsheng.yue,jianyi001,ccloy}@ntu.edu.sg

## 1 Mathematical Details

- **Derivation of Eq. (2)**: According to the transition distribution of Eq. (1) of our manuscript, $\boldsymbol{x}_t$ can be sampled via the following reparameterization trick:

$$\boldsymbol{x}_t = \boldsymbol{x}_{t-1} + \alpha_t \boldsymbol{e}_0 + \kappa\sqrt{\alpha_t}\boldsymbol{\xi}_t, \tag{A.1}$$

where $\boldsymbol{\xi}_t \sim \mathcal{N}(\boldsymbol{x}|0, \boldsymbol{I})$, $\alpha_t = \eta_t - \eta_{t-1}$ for $t > 1$ and $\alpha_1 = \eta_1$.
Applying this sampling trick recursively, we can build up the relation between $\boldsymbol{x}_t$ and $\boldsymbol{x}_0$ as follows:

$$\begin{aligned}
\boldsymbol{x}_t &= \boldsymbol{x}_0 + \sum_{i=1}^{t}\alpha_i\boldsymbol{e}_0 + \kappa\sum_{i=1}^{t}\sqrt{\alpha_i}\boldsymbol{\xi}_i \\
&= \boldsymbol{x}_0 + \eta_t\boldsymbol{e}_0 + \kappa\sum_{i=1}^{t}\sqrt{\alpha_i}\boldsymbol{\xi}_i,
\end{aligned} \tag{A.2}$$

where $\boldsymbol{\xi}_i \sim \mathcal{N}(\boldsymbol{x}|0, \boldsymbol{I})$.
We can further merge $\boldsymbol{\xi}_1, \boldsymbol{\xi}_2, \cdots, \boldsymbol{\xi}_t$ and simplify Eq. (A.2) as follows:

$$\boldsymbol{x}_t = \boldsymbol{x}_0 + \eta_t\boldsymbol{e}_0 + \kappa\sqrt{\eta_t}\boldsymbol{\xi_t}. \tag{A.3}$$

Then the marginal distribution of Eq. (2) in the main text is obtained based on Eq. (A.3).

- **Derivation of Eq. (6)**: According to Bayes's theorem, we have

$$q(\boldsymbol{x}_{t-1}|\boldsymbol{x}_t, \boldsymbol{x}_0, \boldsymbol{y}_0) \propto q(\boldsymbol{x}_t|\boldsymbol{x}_{t-1}, \boldsymbol{y}_0)q(\boldsymbol{x_{t-1}}|\boldsymbol{x}_0, \boldsymbol{y}_0), \tag{A.4}$$

where

$$\begin{aligned}
q(\boldsymbol{x}_t|\boldsymbol{x}_{t-1}, \boldsymbol{y}_0) &= \mathcal{N}(\boldsymbol{x}_t; \boldsymbol{x}_{t-1} + \alpha_t\boldsymbol{e}_0, \kappa^2\alpha_t\boldsymbol{I}), \\
q(\boldsymbol{x_{t-1}}|\boldsymbol{x}_0, \boldsymbol{y}_0) &= \mathcal{N}(\boldsymbol{x}_{t-1}; \boldsymbol{x}_0 + \eta_{t-1}\boldsymbol{e}_0, \kappa^2\eta_{t-1}\boldsymbol{I}).
\end{aligned} \tag{A.5}$$

We now focus on the quadratic form in the exponent of $q(\boldsymbol{x}_{t-1}|\boldsymbol{x}_t, \boldsymbol{x}_0, \boldsymbol{y}_0)$, namely,

$$\begin{aligned}
&-\frac{(\boldsymbol{x}_t - \boldsymbol{x}_{t-1} - \alpha_t\boldsymbol{e}_0)(\boldsymbol{x}_t - \boldsymbol{x}_{t-1} - \alpha_t\boldsymbol{e}_0)^T}{2\kappa^2\alpha_t} - \frac{(\boldsymbol{x}_{t-1} - \boldsymbol{x}_0 - \eta_{t-1}\boldsymbol{e}_0)(\boldsymbol{x}_{t-1} - \boldsymbol{x}_0 - \eta_{t-1}\boldsymbol{e}_0)^T}{2\kappa^2\eta_{t-1}} \\
&= -\frac{1}{2}\left[\frac{1}{\kappa^2\alpha_t} + \frac{1}{\kappa^2\eta_{t-1}}\right]\boldsymbol{x}_{t-1}\boldsymbol{x}_{t-1}^T + \left[\frac{\boldsymbol{x}_t - \alpha_t\boldsymbol{e}_0}{\kappa^2\alpha_t} + \frac{\boldsymbol{x}_0 + \eta_{t-1}\boldsymbol{e}_0}{\kappa^2\eta_{t-1}}\right]\boldsymbol{x}_{t-1}^T + \text{const} \\
&= -\frac{(\boldsymbol{x}_{t-1} - \boldsymbol{\mu})(\boldsymbol{x}_{t-1} - \boldsymbol{\mu})^T}{2\lambda^2} + \text{const}
\end{aligned} \tag{A.6}$$

where

$$\boldsymbol{\mu} = \frac{\eta_{t-1}}{\eta_t}\boldsymbol{x}_t + \frac{\alpha_t}{\eta_t}\boldsymbol{x}_0, \ \lambda^2 = \kappa^2\frac{\eta_{t-1}}{\eta_t}\alpha_t, \tag{A.7}$$

and const denotes the item that is independent of $\boldsymbol{x}_{t-1}$. This quadratic form induces the Gaussian distribution of Eq. (6) in our manuscript.

37th Conference on Neural Information Processing Systems (NeurIPS 2023).

Table 1: Quantitative comparison of different methods on the task of x4 (64→256) bicubic SR . We mark the number of sampling steps for each method by the format of "method-steps".

| Methods | PSNR↑ | SSIM↑ | LPIPS↓ | CLIPIQA↑ | MUSIQ↑ | # Parameters (M) | Runtime (s) |
|---|---|---|---|---|---|---|---|
| IRSDE-100 [4] | 24.48 | 0.602 | 0.304 | 0.513 | 45.382 | 137.2 | 5.927 |
| DDRM-15 [2] | 25.56 | 0.674 | 0.471 | 0.372 | 24.746 | 552.8 | 1.184 |
| I2SB-15 [3] | 26.76 | 0.730 | 0.206 | 0.489 | 53.936 | 552.8 | 1.832 |
| *ResShift*-15 | 26.73 | 0.736 | 0.126 | 0.683 | 58.067 | 121.3 | 0.105 |

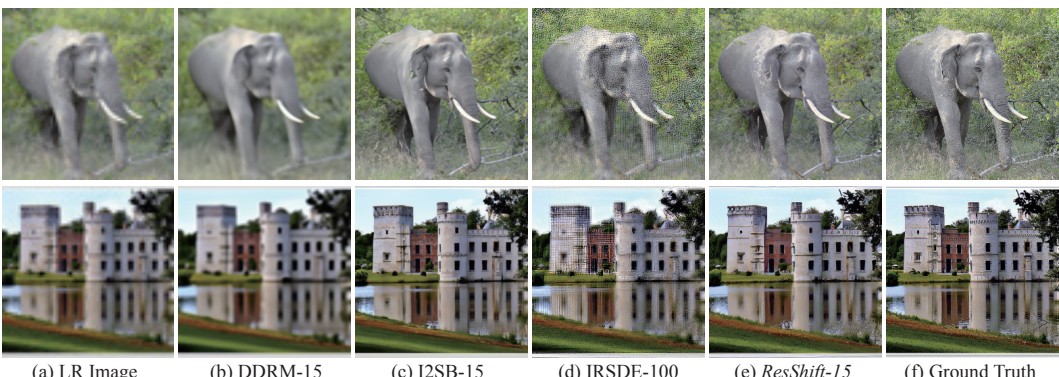

| (a) LR Image | (b) DDRM-15 | (c) I2SB-15 | (d) IRSDE-100 | (e) *ResShift-15* | (f) Ground Truth |
|---|---|---|---|---|---|

Figure 1: Visual comparisons of different methods on the task of x4 (64→256) bicubic super-resolution. Please zoom in for a better view.

## 2 Experiment

### 2.1 Degradation Settings of the Synthetic Dataset

We synthesize the testing dataset *ImageNet-Test* based on the degradation model in RealESRGAN [6] but removing the second-order operation. We observed that the LR image generated by the pipeline with second-order degradation exhibited significantly more pronounced corruption than most of the real-world LR images, we thus discarded the second-order operation to align the authentic degradation better. Next, we gave the detailed configuration on the blurring kernel, downsampling operator, and noise types.

**Blurring kernel.** The blurring kernel is randomly sampled from the isotropic Gaussian and anisotropic Gaussian kernels with a probability of [0.6, 0.4]. The window size of the kernel is set to 13. For isotropic Gaussian kernel, the kernel width is uniformly sampled from [0.2, 0.8]. For an anisotropic Gaussian kernel, the kernel widths along $x$-axis and $y$-axis are both randomly sampled from [0.2, 0.8].

**Downampling.** We downsample the image using the "interpolate" function of PyTorch [5]. The interpolation mode is random selected from "area", "bilinear", and "bicubic".

**Noise.** We first added Gaussian and Poisson noise with a probability of [0.5, 0.5]. For Gaussian noise, the noise level is randomly chosen from [1,15]. For Poisson noise, we set the scale parameter in [0.05, 0.3]. Finally, the noisy image is further compressed using JPEG with a quality factor ranged in [70, 95].

### 2.2 Evaluation on Bicubic Degradation

In this section, we conduct an evaluation of the proposed *ResShift* on the task of x4 (64→256) bicubic SR, against three recent diffusion-based methods, including DDRM [2], IRSDE [4], and I2SB [3]. To ensure a fair comparison, we retrain a new model specifically tailored for bicubic degradation, given that all three comparison methods were originally designed or trained under the same degradation setting. Our testing dataset comprises 3,000 images randomly selected from the validation dataset of ImageNet [1]. During the evaluation phase, we expedite the inference process to 15 steps using the default sampler for both I2SB and DDRM. However, it is worth noting that we retained the sampling steps of 100 for IRSDE, consistent with its configuration in training, since we empirically found that accelerating the inference of IRSDE led to a severe performance drop.

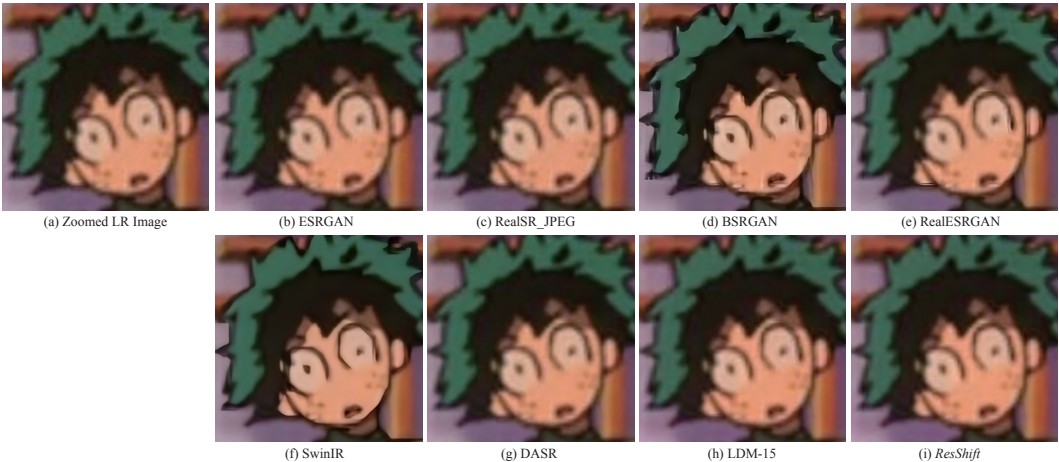

Figure 2: One typical failed case in the real-world dataset of *RealSet65*.

The quantitative comparison results are listed in Table 1, revealing a conspicuous superiority of our proposed *ResShift* beyond the alternative methods across various assessment metrics, parameter counts, and inference throughput. This performance difference underscores the effectiveness and efficiency of the proposed diffusion model. The visual comparison of two exemplar cases is depicted in Fig. 1. Our approach demonstrates superior capability in recovering rich and realistic image details, consistent with the quantitative observations.

## 2.3 Limitation

Albeit its overall strong performance, the proposed *ResShift* occasionally exhibits failures. One such instance is illustrated in Figure 2, where it cannot produce satisfactory results for a severely degraded comic image. It should be noted that other comparison methods also struggle to address this particular example. This is not an unexpected outcome as most modern SR methods are trained on synthetic datasets simulated by manually assumed degradation models [7, 6], which still cannot cover the full range of complicated real degradation types. Therefore, developing a more practical degradation model for SR is an essential avenue for future research.

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

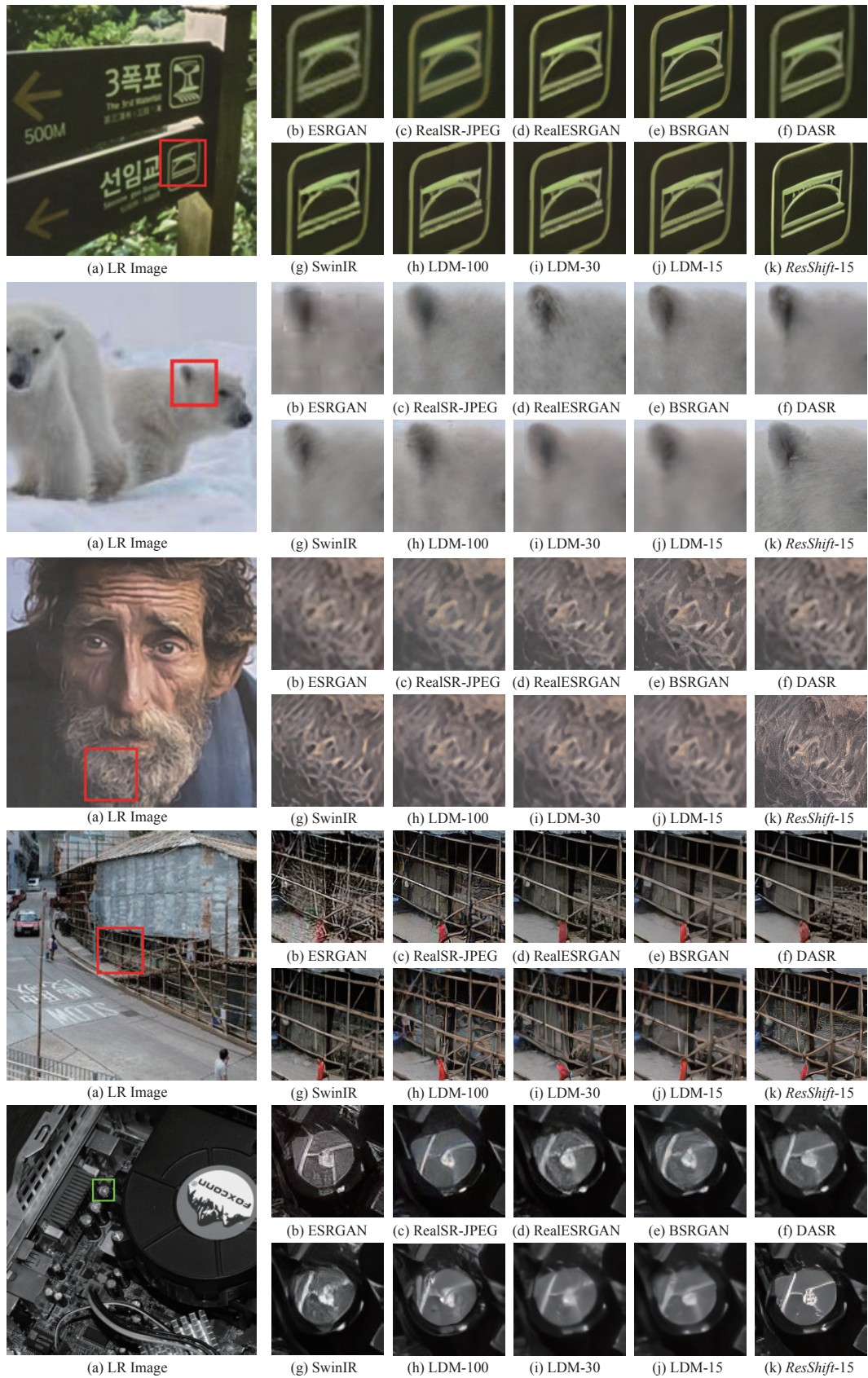

Figure 3: Qualitative comparisons of different methods on real-world datasets. Please zoom in for a better view.

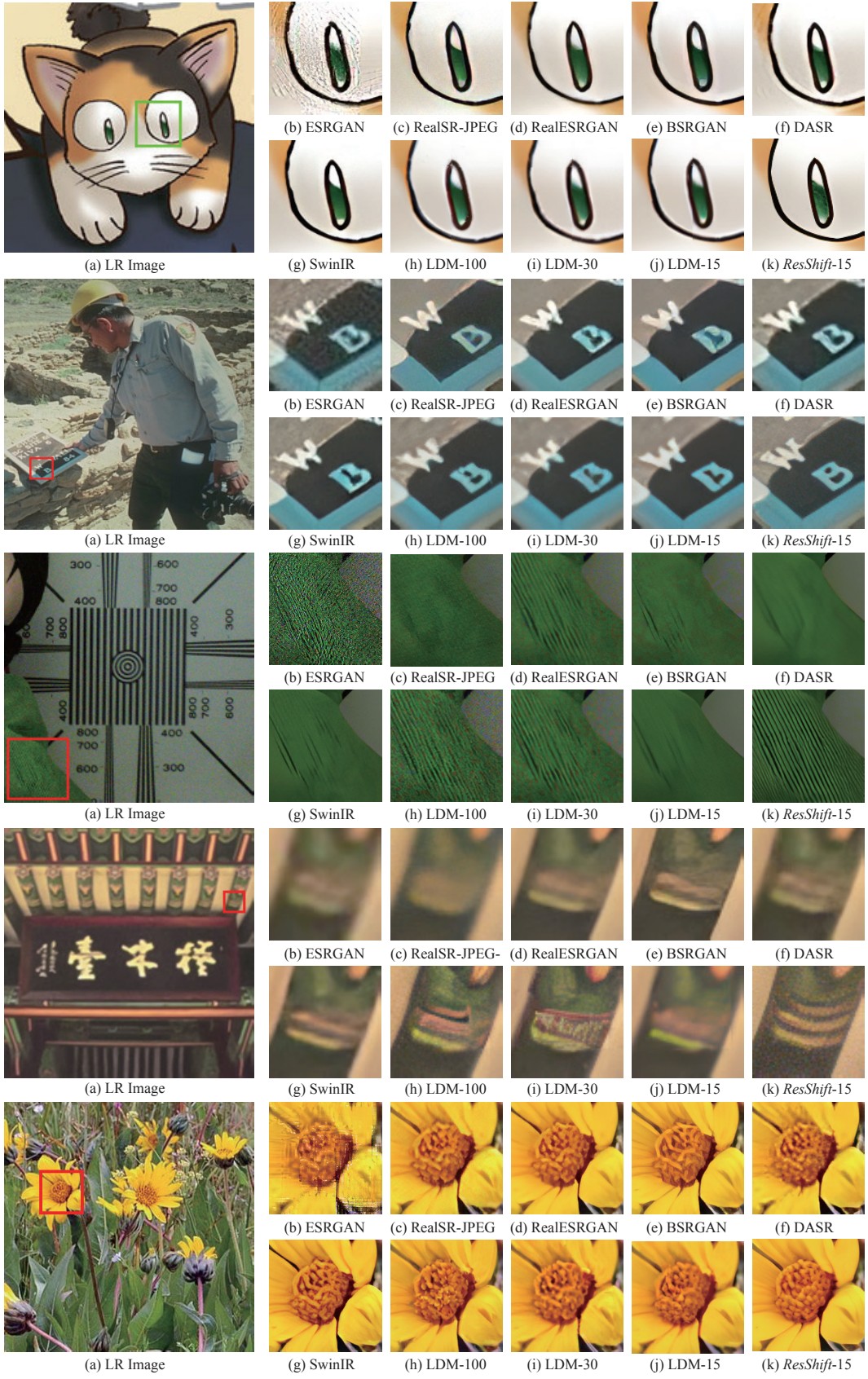

Figure 4: Qualitative comparisons of different methods on real-world datasets. Please zoom in for a better view.