# OpenReview forum: "ResShift: Efficient Diffusion Model for Image Super-resolution by Residual Shifting"
_NeurIPS.cc/2023/Conference — NeurIPS 2023 spotlight_

### Official Review · Reviewer_PfAe · 2023-07-07

**Soundness:** 4 excellent
**Presentation:** 3 good
**Contribution:** 3 good
**Rating:** 6
**Confidence:** 3

**Summary:**

This paper constructs a Markov chain that transfers between high-resolution and low-resolution by shifting the residual between them. They achieve competitive performance to SOTA methods with only 20 sampling steps.

**Strengths:**

1. The paper structure is clear and the proposed method is well-motivated.
2. It only requires small iterating  steps to achieve competitive performances.

**Weaknesses:**

1. A brief figure to introduce Resshift should be provided.
2.  Why this method is called ResShift? The motivation and function should be discussed. Moreover, the ablation study of the res-shift should be provided.

**Questions:**

1. Since there are many math equations, it may be difficult for other researchers to reproduce the method. Would the implementation code be released?

**Limitations:**

The authors have addressed the limitations.

---

> ### Author Rebuttal · Authors · 2023-08-09
>
> > **Q1. A brief figure to introduce ResShift should be provided.**
>
> Thanks for your suggestion. We have provided an overview of our model in Fig 3. (a) in the attached rebuttal document, and will add it to our paper in the revised version.
>
> > **Q2. Why this method is called ResShift? The motivation and function should be discussed.**
>
> The motivation of this work is derived from an intuitive observation, which posits that the transition from an high-resolution (HR) image to its low-resolution (LR) counterpart is more efficient compared to that from the HR image to a Gaussian noise. This assertion stems from the fact that the residual between the HR-LR image pairs are often small. Therefore, this study develops a novel diffusion model that builds up a Markov chain between the HR-LR images by gradually **Shift**ing the **Res**idual of them. We thus refer to the proposed diffusion model as ResShift, encapsulating the essence of this methodology.
>
> > **Q3. Moreover, the ablation study of the res-shift should be provided.**
>
> The whole formulation of our proposed model is methodically crafted based on the strategy of residual shifting. It is thus impossible to make an ablation study on the res-shift. However, we have provided sufficient ablation analysis on the other configurations of our model in Sec. 4.2.
>
> > **Q4. Would the implementation code be released?**
>
> We promise to release the source code of this work after revision.

---

> > ### Comment · Reviewer_PfAe · 2023-08-18
> >
> > I would like to keep my rating as "Weak Accept"

---

### Official Review · Reviewer_k6dY · 2023-07-07

**Soundness:** 3 good
**Presentation:** 3 good
**Contribution:** 3 good
**Rating:** 6
**Confidence:** 4

**Summary:**

This paper studies diffusion-based image super-resolution (SR) with the goal of reducing the number of diffusion steps. The key intuition is to only learn the residual between an LR-HR image pair, thereby shortening the diffusion path. To this end, the paper introduces ResShift, a novel diffusion framework where the mean of the prior distribution is the LR image. It further provides a derivation of the forward and reverse processes, identifies an effective parameterization of the denoiser network, and analyzes the fidelity-realism tradeoff induced by the noise schedule. Extensive quantitative and qualitative experiments are conducted to verify the effectiveness and efficiency of the proposed method. In particular, ResShift achieves competitive results on the 4x SR task at a low cost of 20 diffusion steps, outperforming the strong baseline of LDM under the same sampling budget.

**Strengths:**

- The paper makes the key observation that the residual between an LR-HR image pair is often small. It builds a novel diffusion framework around this insight to reduce the number of sampling steps for improved efficiency. Incorporating domain knowledge into the design of a diffusion model is a key strength of the paper.

- Within the proposed diffusion framework, the paper studies the fidelity-realism tradeoff induced by noise schedules through detailed theoretical analysis and ablation experiments.

- The proposed method is evaluated on both synthetic and real datasets. It demonstrates comparable or sometimes better results with respect to strong baselines. In particular, it consistently outperforms LDM under the same sampling budget (20 or 40 steps).

- Overall, the paper is well-written, and the flow of presentation is easy to follow.

**Weaknesses:**

- My main concern is that the comparison with LDM is unfair. In the main experiments, the number of diffusion steps is limited to 20 or 40. This makes sense, as the goal is to demonstrate that ResShift yields stronger results under a tight sampling budget. However, LDM is put at a significant disadvantage in this comparison, because its default DDIM sampler is not well-suited for short sampling paths. In this regime, more powerful samplers exist (e.g., Heun and DPM solvers) and, according to the reviewer's own experience, will likely produce far better results within 20 or 40 NFEs. This raises an important question: is the proposed diffusion model tailored for SR truly superior to the vanilla model in terms of efficiency, when the latter is paired with a fast solver? This is key to assessing the paper's contribution, yet is not addressed by the current set of experiments.

- DDRM is another diffusion-based model for SR in addition to LDM. The authors are encouraged to compare ResShift against DDRM as well.

- The qualitative results on real data seem inconsistent across different images. In Figure 4, ResShift produces sharper and more detailed output than the baselines, whereas it is more prone to artifacts as shown in Figure 1 in the supplement. I am wondering what causes this discrepancy. Are the images from different real datasets, or is there just a lot of randomness?

**Questions:**

Please see the section above for questions.

**Limitations:**

The paper discusses limitations of the method. The authors are encouraged to provide a short discussion on the potential negative societal impact of their work, such as hallucination of inappropriate image content.

---

> ### Author Rebuttal · Authors · 2023-08-09
>
> > **Q1. The comparison with LDM is unfair. More advanced samplers should be considered.**
>
> As suggest, we conduct a comparison to LDM with 20 sampling steps accelerated by more advanced samplers, including PNDM (ICLR 2022) and DPM (NeurIPS 2022). The quantitative comparison results on the testing dataset of ImageNet-Test are presented as follows:
>
> | Methods | Sampler |PSNR$\uparrow$ | SSIM$\uparrow$   | LPIPS$\downarrow$ | CLIPIQA$\uparrow$ | MUSIQ$\uparrow$ |
> | :----:        | :----:      | :----:   | :----:   |:----:    | :----:       | :----:     |
> | LDM-20  | DDIM     | 24.76  | 0.639  |0.284  | 0.630      | 48.355   |
> | LDM-20  | PNDM   | 22.87  | 0.549   |0.285  | 0.738      | 55.820  |
> | LDM-20  | DPM      | 22.28  | 0.526   |0.302  | 0.728      | 54.123  |
> | F-ResShift-20  | -   |23.72   | 0.615   |0.246  | 0.773      | 57.408  |
>
> It can be easily seen that the incorporation of these advanced sampling techniques brings up pronounced enhancement in perceptual quality, as assessed by CLIPIQA and MUSIQ. However, this performance augmentation comes at a substantial deterioration on fidelity, quantified by PSNR and SSIM. Even in light of this, our proposed method still consistently exhibits evident superiorities across all five metrics. Therefore, the investigations undertaken in this study assume a pivotal and noteworthy significance.
>
> > **Q2. Performance comparison with DDRM.**
>
> DDRM is a non-blind diffusion-base restoration method, and cannot handle the complicated real-world image super-resolution with unknown degradation types. Thus, we omit the comparison with DDRM in the submitted manuscript. To more comprehensively evaluate the proposed method, we conduct a fair comparison with DDRM on the task of x4 bicubic super-resolution. Please see Q1 of the global response.
>
> > **Q3. Artifacts in Fig. 1 of the supplement.  Is there just a lot of randomness?**
>
> We want to elucidate that the artifacts observed in rare instances,  e.g., the second example in Fig. 1 of the supplement,  can be predominantly attributed to the inadequate
>  training. Our model in the submitted version undergoes a training over 500k iterations with a batch size of 64. We empirically find that the artifacts can be substantially mitigated by prolonging the training process to 800k iterations. This extended model is denoted as F-Reshift+, and the revised results are presented in Fig. 4 of the accompanying rebuttal file. Notably, it is necessary to note that LDM is trained with 2560k iteration in a batch size of 256, incurring significantly greater computational resources than our proposed approach.
>
> Besides, the restored results by F-ResShift+ under multiple different random seeds are also shown in Fig. 4 (h)-(k). We can see that the randomness among multiple solutions is exceedingly slight, thus deemed acceptable in the task of image super-resolution.

---

> ### Comment · Reviewer_k6dY · 2023-08-15
> **Updated rating**
>
> The rebuttal addressed my concerns. I thus raised my rating from borderline reject to weak accept. The authors are encouraged to include quantitative and qualitative results of LDM with different solvers in their revision.

---

> > ### Author Response · Authors · 2023-08-15
> >
> > We're happy that the rebuttal have addressed your concerns. As suggested, we will include the quantitative and qualitative results of LDM with various solvers in our revised version.
> >
> > Thanks for your feedback.

---

### Official Review · Reviewer_Mv36 · 2023-07-12

**Soundness:** 3 good
**Presentation:** 1 poor
**Contribution:** 3 good
**Rating:** 5
**Confidence:** 5

**Summary:**

The paper presents a new image super-resolution diffusion model called "resshift", which aims to address the efficiency issue in diffusion models. Existing acceleration strategies often yield over-smooth results. To combat this, the authors propose a new diffusion model for super-resolution that can produce favorable results in just 20 sampling steps. This method is based on operating on residuals and the paper also proposes a noise-adding strategy to aid this process. The main experiments are performed on real super-resolution settings, imitating the prior setup of Real ESRGAN, and includes testing on real images.

**Strengths:**

The experimental results appear quite promising. Specifically, the outcomes produced with 20 steps seem slightly better than existing methods. This shows that the proposed method can efficiently achieve super-resolution with fewer sampling steps, and the introduced noise-adding strategy appears to be a beneficial supplement to the overall approach.

**Weaknesses:**

The major issue with this paper lies in its presentation. The central idea of the paper is not clearly articulated. In the second section, although the method is described meticulously, it does not seem significantly different from existing methods. The authors do not clearly highlight what distinguishes their approach. It is recommended that the authors construct a table to illustrate why the "shift" and "residual" ideas are useful, and provide a conceptual narrative at the beginning of the second chapter to give readers an overall impression before diving into mathematical expressions.


Secondly, Figure 2 could potentially be misleading. It should be the image obtained by adding noise in the VQ-VAE's latent space, but this is not just noise because a conversion to RGB image is performed. This needs further clarification since it's difficult to discern that Gaussian noise is added without any explanation.


Lastly, the method's versatility needs to be elaborated on. The method does not seem to be confined to super-resolution and appears to be applicable to all image restoration models. The question arises whether it could be applied to other image processing tasks. There also needs to be more comparative analysis, as many recent acceleration methods, including CM models, have not been included in the comparison.


**Questions:**

see Weakness

**Limitations:**

Yes

---

> ### Author Rebuttal · Authors · 2023-08-09
>
> > **Q1. The major issue with this paper lies in its presentation. The central idea of the paper is not clearly articulated. In the second section, although the method is described meticulously, it does not seem significantly different from existing methods. The authors do not clearly highlight what distinguishes their approach.**
>
> The motivation of this work is derived from an intuitive observation, which posits that the transition from an high-resolution (HR) image to its low-resolution (LR) counterpart displays enhanced efficiency, characterized by a reduced number of diffusion steps, in comparison to the transition from the HR image to a Gaussian noise. This assertion stems from the fact that the residual between the HR-LR image pairs are often small. Such a diffusion process can be implemented by gradually shifting this residual, and one corresponding visualization of the hidden states during transition is shown in Fig. 3(a) of the affiliated rebuttal file. The experimental results also well substantiate the rationality of this motivation. Besides, we are very happy on that our motivation and the presentation are highly recognized and appreciated by all the other Reviewers.
>
> Furthermore, we want to kindly clarify the contribution of this work so as to distinguish from existing methods. Based on the aforementioned motivation, we innovate by designing a diffusion model from the mathematical formulation of the forward and reverse processes to the optimization strategy. The devised model enables a seamless transition between the HR-LR image pairs, being different from extant approaches that transition from HR image to Gaussian noise. Beyond the scope of this theoretical model innovation, a more flexible noise schedule is also proposed to better control the perception-distortion trade-off. Collectively, these contributions serve to establish clear differentiation between our approach and previous research.
>
> > **Q2. Figure 2 could potentially be misleading. It should be the image obtained by adding noise in the VQ-VAE's latent space, but this is not just noise because a conversion to RGB image is performed. This needs further clarification since it's difficult to discern that Gaussian noise is added without any explanation.**
>
> We sincerely appreciate this constructive suggestion. We want to firstly clarify that our proposed method provides a general framework to establish a transition between any two variables, i.e., the HR image $x_0$ and LR image $y_0$ in the context of image super-resolution. Given a pre-trained VQGAN with encoder $E$ and decode $D$, it is naturally to extend our method to model the transition between $z_0$ and $v_0$ in latent space, where $z_0=E(x_0 )$, $v_0=E(y_0)$. It should be noted that when modelling in latent space, the encoding and decoding procedures only occur once in the initial and terminal states during forward and reverse process. For example, in the reverse process, we only need to estimate $z_0$ from $v_0$ that encoded from $y_0$ using the trained diffusion model, and then obtain the restored $x_0$ via $x=D(z_0)$. Thus, it is unnecessary to concern the conversion between the latent space and the RGB space for the introduced Gaussian noise in our model.
>
> Next, we give more details about the visualization in Fig. 2. Specifically, we first mapped $x_0$ and $y_0$ to $z_0$ and $v_0$ via the encoder $E$, and then generate the intermediate states $\lbrace z_t \rbrace_{t=1}^T$ using the transition kernel in Eq. (1) in latent space, and finally convert them to the RGB space through $x_t=D(z_t)$. In fact, Figure 2 in our paper visualizes the decoded image $x_t$. We will add this explanation in our revised version for easy understanding.
>
> > **Q3. Whether the proposed method could be applied to other image processing tasks.**
>
> Yes. The proposed method is a general methodology to address a wide spectrum of restoration tasks. To verify such a versatility of our method, we further evaluate its effectiveness on the task of blind face restoration. A subset of 3000 images are randomly selected from CelebA as our testing dataset following CodeFormer (NeurIPS 2022). The quantitative comparisons to GFPGAN (CVPR 2021), VQFR (ECCV 2022), and CodeFormer are listed in Table 3 of the associated rebuttal document. Furthermore, the qualitative comparison on one real-world example is also shown in Fig. 1. We can easily see that ResShift achieves the best or at least comparable performance to recent SotA method CodeFormer. This compellingly suggests the potential for the seamless extension of our method to blind face restoration.

---

### Official Review · Reviewer_RmTK · 2023-07-22

**Soundness:** 3 good
**Presentation:** 3 good
**Contribution:** 2 fair
**Rating:** 6
**Confidence:** 4

**Summary:**

The authors propose a new diffusion model, based on the principle of residual shifting, that is able to converge to a good looking image in a low number of diffusion steps, improving in terms of high resolution image inference time at least over the Latent Diffusion Model (LDM) SR variant, another well-established diffusion based solution  for image upscaling.

The authors provide extensive ablations (both is the main manuscript and the supplementary material) showing that the model is able to achieve a significant level of performance, at least in terms of subjective results analysis. The trade-off between the number of diffusion steps and the achieved "image quality" (subjective) is well explored. However, some concerns and comments will be found in the appropriate section of the review.

 The metrics chosen for performance quantification (PSNR, SSIM, LPIPS, CLIPIQA,  MUSIQ) show mixed results, with the model characterized by SOTA performance in terms of non-reference image metrics, with strong indicators in terms of CLIPQA and MUSIQ. In terms of LPIPS, the model also shows an advantage over the compared methods. In terms of PSNR and SSIM, the model shows its limitations, however explainable, given the nature of the algorithm and the followed framework.

**Strengths:**

1) The paper is generally well written and easy to follow. The provided details are enough to offer a good overview of the proposed method.
2) The authors provide ablations showcasing the influence of multiple parameters characterizing the proposed method.
3) The quality of the results selected for visual comparison matches the performance level in terms of quality assessment related metrics.
4) An advantage over the LDM in terms of inference time/diffusion steps needed  to achieve a similar image quality assessment performance can be easily derived from the Table 2 of the main manuscript.
The quality of the results provided for visual comparison and the perspective for more efficient diffusion models explain my rating.

**Weaknesses:**

1) The efficiency claim made in the title is not clearly supported by the provided evaluations. Even if the advantage over LDM in terms of inference time and diffusion steps needed is clear, there is no comparison with the other methods considered for comparison (Table 3 and 4).
This work should clarify the question mark over the opportunity of using a diffusion model for Efficient Image Super Resolution, show the model strengths and let the readers evaluate the trade-off between reconstruction fidelity and subjective "image quality". This is why I added this as the first weak point of the authors claim.
2) The choice in presenting the results of their ablative study (Table 1) versus the against-SOTA (Table 3,4) is somehow strange. the configurations in terms of data are different making it impossible to quantify the advantage between their proposed models and different configurations of LDM. Also, a fully supervised Image Super Resolution CNN architecture could be also added, to better support the efficiency claim as performance gain per inference-time millisecond.
3) A certain trend can be observed in the first 5 lines of the Table 1, where the performance of the model in terms of reconstruction fidelity decreases with the number of diffusion steps  followed. This can show an overfitting behaviour on some properties of the training set that are not aligned with the properties of the testing split.
       3.1)  Have the authors focused their efforts on strategies to supervise the training of their model in order to retain as much as possible
       from the reconstruction fidelity, while achieving significant subjective performance?
4) When setting the numbers of diffusion steps to 20, there seems to be a strong correlation between the parameter p value and the
reconstruction fidelity of the model. However, the value for p chosen for the final configuration was 0.8. Was the choice made given
the impact in performance in terms of CLIPQA/MUSIQ? Why?

5) A comparison with another referenced method claiming improved inference time for a diffusion-based algorithm (maybe also Reffusion) would be useful for the potential reader, to understand the advantages of the proposed method (reference 48 of the main manuscript).


**Questions:**

1) Where is the difference  in the number of parameters of the diffusion backbone against the LDM coming from?

---

> ### Author Rebuttal · Authors · 2023-08-09
>
> > **Q1. Efficiency comparison with the other methods in Table 3 and 4.**
>
> We have offered more comprehensive comparable analysis on the efficiency as suggested. Please see Q2 of the global response.
>
> > **Q2. The choice in presenting the results of their ablative study (Table 1) versus the against-SOTA (Table 3,4) is somehow strange. the configurations in terms of data are different making it impossible to quantify the advantage between their proposed models and different configurations of LDM.**
>
> We would like to elucidate that the ablative study in Table 1 and the comparisons with current SotA methods in Table 3 are both conducted on the synthetic testing dataset “ImageNet-Test” (see Sec 4.1 of our manuscript). It is imperative to note that the dataset configuration remains consistent between Table 1 and Table 3. To better evaluate the effectiveness of our method in authentic real-world scenarios, we further made a comparison with existing method on three real-world datasets and presented the results in Table 4. Due to the unavailability of ground truth images in real-world datasets, our scrutiny primarily focused on non-reference metrics, namely CLIPIQA and MUSIQ, as highlighted in Table 4.
>
> > **Q3. A certain trend can be observed in the first 5 lines of the Table 1, where the performance of the model in terms of reconstruction fidelity decreases with the number of diffusion steps followed. This can show an overfitting behaviour on some properties of the training set that are not aligned with the properties of the testing split. 3.1) Have the authors focused their efforts on strategies to supervise the training of their model in order to retain as much as possible from the reconstruction fidelity, while achieving significant subjective performance?**
>
> We kindly argue that the decline in reconstruction fidelity with an increase in diffusion steps should not be attributed to an overfitting onto the training data. Rather, this is a well-known phenomenon called “Perception-distortion Trade-off” [1] in the field of image restoration. In particular, the augmentation of the generative capability of a restoration model, such as elevating the sampling steps for a diffusion-based method or amplifying the weight of the adversarial loss for a GAN-based method, will result in a deterioration in fidelity preservation, while concurrently enhance the authenticity of restored images. That’s mainly because the restoration model with powerful generation capability tends to hallucinate more high-frequency image structures, thereby deviating from the underlying ground truth.
>
> To facilitate a more comprehensive comparison between our method and LDM, we have plotted the perception-distortion curves of them in Fig. 3 (b) of the accompanying rebuttal file. Herein, the perception and distortion are measured by the metrics of CLIPIQA and mean square-error (MSE), respectively. The plot reflects the perception quality and the reconstruction fidelity of the proposed method and LDM across varying numbers of diffusion steps, i.e,, 10,  20, 30, 40, 50, and 100. Significantly, the perception-distortion curve of our ResShift consistently resides beneath that of the LDM, thereby indicating its superior capacity to manage the perception-distortion equilibrium.
>
> [1] The Perception-Distortion Tradeoff, CVPR 2018.
>
> > **Q4. When setting the numbers of diffusion steps to 20, there seems to be a strong correlation between the parameter $p$ value and the reconstruction fidelity of the model. However, the value for p chosen for the final configuration was 0.3. Was the choice made given the impact in performance in terms of CLIPQA/MUSIQ? Why?**
>
> Yes, you’re right. In our final model, the hyper-parameter of $p$ is set as 0.3, hoping to enhance the perception quality measured by LPIPS, CLIPIQA, and MUSIQ. In the real applications of super-resolution, the prevailing preference consistently gravitates towards outcomes characterized by heightened perceptual quality, in contrast to the conventional emphasis on reconstruction fidelity. This principled inclination is grounded in one common observation that the restored image with superior reconstruction fidelity, as measured by the metric of PSNR, often tends to be blurry. Actually, one can adjust the hyper-parameter $p$ freely according to your requirement.
>
> > **Q5. Performance comparison with the other reference methods.**
>
> Please see the Q1 of the global response.
>
> > **Q6. Where is the difference in the number of parameters of the diffusion backbone against the LDM coming from?**
>
> Regarding the diffusion backbone, we adopted the codebase of the guided diffusion model (https://github.com/openai/guided-diffusion) and followed its default configurations. The difference in the number of parameters arises from different condition manners on the time embedding within the block of ResNet. In the guided diffusion model, a spatial feature transform layer is employed to modulate the features of ResNet based on the time embedding. In contrast, the time embedding vector is directly added to the features of ResNet in LDM.
>
> It should be noted that we have not specifically optimized the backbone to pursue more performance gain, just following the settings in existing work. The superiority of our method primarily comes from the elaborate design on the diffusion model, which serves as the core contribution of this work.

---

> > ### Comment · Reviewer_RmTK · 2023-08-15
> > **Updated Rating**
> >
> > Since some of the concerns were addressed by the authors in their rebuttal, there is a clear advantage of the method against the compared LDM in terms of performance, while still lagging behind other GAN based approaches. Authors are encouraged to clearly describe their residual shift procedure, to emphasize their contribution, and  also the novelty of their method. The evaluations provided in the rebuttal clearly would help the reader to understand the advantages of their method and its potential. Thus, I am considering a "Weak accept" rating for the submission, after the rebuttal.

---

> > > ### Author Response · Authors · 2023-08-15
> > > **Response to the Reviewer's feedback**
> > >
> > > Thanks for your feedback. As suggested, we will clearly describe the residual shift procedure, the contribution and the novelty of this work, and some evaluations in the rebuttal to our revised version.
> > >
> > > According to the response, you tend to increase the rating to "weak accept". We thus want to kindly note that whether you forget to change the rating in the system?
> > >
> > > Thanks again!

---

### Official Review · Reviewer_vFLR · 2023-07-23

**Soundness:** 3 good
**Presentation:** 3 good
**Contribution:** 3 good
**Rating:** 6
**Confidence:** 4

**Summary:**

This paper introduces a diffusion-based image super-resolution method.
The proposed method starts to generate an HR image from a given LR image directly (learns to generate the residual image between LR and HR), not start from a noise.
Therefore, the method can generate the output image faster than previous works, specifically, with only 20 sampling steps.
In addition, there are some hyperparameters to control the fidelity-perceptual quality trade-off of output images, namely, T, p, and k.

**Strengths:**

- The idea of starting from LR image and utilizing a residual image is proper to the SR task and this is different from conditioning the LR image.
- This successfully makes the method takes only short steps in the inference using a comparable number of parameters and runtime compared to previous method LDM.
- In addition, it is practical for users to control the fidelity-perceptual quality trade-off of the SR model by adjusting the hyper-parameters.
- The paper shows the superiority in terms of perceptual quality (LPIPS, CLIPIQA, MUSIQ) in synthetically degraded images and real images as well.

**Weaknesses:**

- There is no visual comparison on synthetic SR testsets so it is hard to fully understand the effectiveness of the method, but the paper argues superiority in this aspect.
- Several previous methods are listed in Sec 3 (Related Work), however, only LDM is compared in experiments. Are there any difficulties
or problems in comparing with other methods?


**Questions:**

- In L87, the image value range of [0,1] but I think the residual image is within [-1,1]. Does the following explanation still hold if I am correct?
- In L135, how this is achieved? I think it is weird because the method does not learn to generate a noise distribution, and what does this description mean?
- It would be better if runtimes/parameters are added for other methods in Table 3 (especially vs GAN-based methods).

**Limitations:**

- Please see the weaknesses and limitations.
- My rating is a reflection of the weaknesses and I look forward to the authors' feedback on the weaknesses.

---

> ### Author Rebuttal · Authors · 2023-08-09
>
> >  **Q1. Visual comparison on the synthetic dataset.**
>
> As suggested, we have presented one visual comparison on the synthetic dataset in Fig. 2 of the associated rebuttal file. Evidently, the proposed method outperforms other competing approaches in terms of both fidelity and realism. In our revised version, we will add more qualitative comparison examples to address this concern.
>
> > **Q2. Performance comparison with other related methods.**
>
> Please see Q1 of the global response.
>
> > **Q3. I think the residual image is within [-1,1]. Does the following explanation still hold if I am correct?**
>
> Yes. The residual image $e_0$ is within [-1,1], and the following explanation still holds on. The shifting sequence $\lbrace\eta_t\rbrace_{t=1}^{T}$ increases monotonically, indicating $\alpha_t = \eta_t -\eta_{t-1} > 0 $ and $\alpha_t<1$, we thus have
>
> $$\text{max}[\alpha_t e_0] < \alpha_t < \sqrt{\alpha_t},$$
>
> where $\text{max}[\cdot]$ is the pixel-wise maximization across all pixels, $e_0 \in R^{h\times w \times c}$, $h$, $w$, and $c$ represent the image height, width, and channels, respectively.
>
> > **Q4. Explanation of Line 135.**
>
> In our devised model, the transition distribution defined in Eq. (1) gradually shifts the residual, accompanied by a minor perturbation by Gaussian noise, during the forward process. The intensity of the Gaussian noise is controlled by the hyper-parameter $\kappa$. For a sufficiently large value of $\kappa$, e.g., 40,  the noise perturbation becomes predominant within the diffusion process, leading to a convergence towards a Gaussian diffusion model.
>
> For a discrete Gaussian diffusion model, the progression of the propagation can be equivalently reflected by the signal-to-noise-ratios of the hidden states (SNRs) in each timestep. By setting $\kappa=40$, $p=0.8$, and $T=1000$, our proposed method demonstrates a closely similar SNR curve to LDM as shown in Fig. 2(g) in our paper. Furthermore, Fig. 2 (e) and (f) visualize the intermediate states of the forward process as observed in our method and LDM. Combining the SNR profile and the visual comparison, it becomes apparent that our method degenerates into a diffusion process that is very close to LDM under such a setting.
>
> > **Q5. Efficiency comparison to the GAN-based.**
>
> Please see Q2 of the global response.

---

> > ### Comment · Reviewer_vFLR · 2023-08-16
> >
> > Thanks for the detailed rebuttal. I checked the visual results and my questions are resolved so I would like to update my rating BR to WA. Please include missing results (regarding Q1, Q2, Q5) and other reviewers' requests as well in the supplementary.

---

### Author Rebuttal · Authors · 2023-08-09

Dear AC and reviewers,

We sincerely thank all reviewers for their constructive comments. Since Reviewer vFLR, Reviewer RmTK, and Reviewer k6dY all concern the performance comparison with other related methods, Reviewer vFLR and Reviewer RmTK both require to supplement the efficiency comparison to GAN-based methods. Thus, we address these two issues here.

>  **Q1. Performance comparison with other related methods.**

Since our proposed method primarily focus on the real-world image super-resolution, we exclusively compare with recent approaches that share a similar orientation. Regarding IRSDE [48] or its enhanced version Refusion (CVPR Workshop 2023), the publicly accessible models are specifically trained for the bicubic super-resolution or stereo super-resolution tasks, and cannot handle the general real-world image super-resolution. Correspondingly, I2SB [49] is similarly limited to bicubic super-resolution. DDRM (NeurIPS 2022) is a non-blind image restoration method, thus lacking the capability to address the real-world image super-resolution with intricate and unknown degradation. For InDI [47], the source code and the pre-trained model are not publicly available. Therefore, our paper omits the comparison to these methods.

Furthermore, we train a model tailored for x4 bicubic super-resolution, thereby facilitating a fair and comparison against IRSDE, I2SB, and DDRM. For I2SB and DDRM, we expedite the inference process to 20 steps using the default sampler. For IRSDE, we retain the sampling steps same with its setting in training, , i.e., 100 steps,  because accelerating this process during inference will yield a severe performance drop. To conduct a comparative analysis, 3,000 images were randomly selected from the validation dataset of ImageNet to serve as our testing dataset. The comprehensive quantitative comparison results are listed in Table 1 of the accompanying rebuttal document. We can easily observe that the proposed method outperforms the aforementioned approaches across various evaluation metrics, the number of parameters, and the inference speed. This indicates the efficacy of our meticulously designed diffusion model.

> **Q2. Efficiency comparisons to other competing methods.**

We provide a comprehensive comparison on the performance and efficiency of our proposed method to LDM and the GAN-based methods in Table 2 of the rebuttal document. For a more holistic assessment, the efficiency evaluations of our method against recent diffusion-based techniques are also presented in Table 1 of the rebuttal document.

Combining these comparative results, we obtain the following conclusions: i)  In contrast to existing diffusion-based methodologies, our proposed method have achieved significant improvement in both performance and efficiency. ii) While current diffusion-based methods have shown notable superiority due to their powerful generation capability, they still lag behind GAN-based models in terms of efficiency. Notably much faster than the diffusion-based techniques as shown in Table 1, our proposed method yet remains twice as slow as SwinIR, the present SotA GAN-based model. We will explore how to further speed up our method in the future work.

---

> ### Author Response · Authors · 2023-08-14
> **Look forward to your feedback.**
>
> Dear AC and Reviewers,
>
> Thanks again for the constructive comments of all the Reviewers. As the deadline for discussion is approaching, we sincerely look forward to your further feedback. Please feel free to let us know if you have any further concerns or comments.
>
> Thanks. The authors.

---

### Decision · Program_Chairs · 2023-09-21

**Decision:**

Accept (spotlight)

**Comment:**

The work proposes an efficient diffusion model for image super-resolution by shifting the residuals and not (re)building the image from scratch like prior work does. This leads to significant benefits as supported by the reported experimental results.

The authors provided satisfactory responses and convinced Reviewers vFLR, RmTK and k6dY to upgrade their ratings. Thus, for this work there is a consensus on the positive side (4xWeak Accept, 1xBorderline Accept) from all five Reviewers.

After reading the paper and carefully checking the reviews and the authors' responses the ACs agree with the reviews that the present work makes important contributions and is of interest for the community.

The authors are invited to further refine their paper for the camera ready by including (part of) information/details from their responses to the reviewers' comments.